# Real-Time and Efficient Multi-Scale Traffic Sign Detection Method for Driverless Cars

**DOI:** 10.3390/s22186930

**Published:** 2022-09-13

**Authors:** Xuan Wang, Jian Guo, Jinglei Yi, Yongchao Song, Jindong Xu, Weiqing Yan, Xin Fu

**Affiliations:** 1School of Computer and Control Engineering, Yantai University, Yantai 264005, China; 2College of Transportation Engineering, Chang’an University, Xi’an 710064, China; 3Engineering Research Center of Highway Infrastructure Digitalization, Ministry of Education, Xi’an 710064, China

**Keywords:** traffic sign detection, attention mechanism, feature pyramid network

## Abstract

Traffic signs detection and recognition is an essential and challenging task for driverless cars. However, the detection of traffic signs in most scenarios belongs to small target detection, and most existing object detection methods show poor performance in these cases, which increases the difficulty of detection. To further improve the accuracy of small object detection for traffic signs, this paper proposed an optimization strategy based on the YOLOv4 network. Firstly, an improved triplet attention mechanism was added to the backbone network. It was combined with optimized weights to make the network focus more on the acquisition of channel and spatial features. Secondly, a bidirectional feature pyramid network (BiFPN) was used in the neck network to enhance feature fusion, which can effectively improve the feature perception field of small objects. The improved model and some state-of-the-art (SOTA) methods were compared on the joint dataset TT100K-COCO. Experimental results show that the enhanced network can achieve 60.4% mAP(Mean Average Precision), surpassing the YOLOv4 by 8% with the same input size. With a larger input size, it can achieve a best performance capability of 66.4% mAP. This work provides a reference for research on obtaining higher accuracy for traffic sign detection in autonomous driving.

## 1. Introduction

With the continuous development of deep learning, computer vision has achieved remarkable achievement and can be used in real-life scenarios. Object detection is one of the current subtasks of computer vision. It is widely used in the field of intelligent transportation, such as motor/non-motor vehicle recognition, pedestrian detection, autonomous driving vehicles, etc. This paper focuses on the study of traffic sign detection, which is a significant part of high-precision map construction for driverless cars. Actual traffic scenes are very complex: traffic signs are densely distributed, and weather factors such as fog, rain, and snow can also affect the accuracy of traffic signs detection. In this case, it is undoubtedly fatal for pedestrians and drivers. Therefore, it is particularly vital to improve the performance of traffic sign detection in various complex environments.

In recent years, computer vision has seen rapid development thanks to the contributions of researchers to convolutional neural networks. Earlier neural networks could only handle classification tasks in small, low-resolution datasets, such as CIFAR [1]. Then, Alex et al. [2] proposed AlexNet, which consists of convolutional and fully connected layers, to achieve the classification task in large datasets such as ImageNet [3]. After that, researchers proposed networks with more layers, such as VGG [4] and GoogLeNet [5], which improved the accuracy of networks to some extent. However, the gradient will drop or disappear when reaching a certain depth. He et al. [6] proposed ResNet, which uses cross-layer connections to fuse the input with the output of the residual blocks, ensuring that deeper network layers can obtain no fewer features than shallower network layers, effectively alleviating the phenomenon of insignificance or even disappearance of deeper features. Most of the previous networks improve the network performance by adjusting the depth, width, and input image resolution of the network model. Tan et al. [7] proposed EfficientNet based on the search for the best scaling factor to unify the three factors, which first established an optimal benchmark and then adjusted the benchmark network based on different scaling factors.

With strong detection capabilities and real-time performance, YOLOv4 has been applied to many scenarios, but it often fails to obtain satisfactory results in traffic sign datasets containing many small objects. Our paper improves the model for the following two problems. Due to the nature of convolutional networks, global features are lost in the feature extraction process, and feature links between dimensions are ignored. Additionally, it is challenging to capture localization information and feature information of small objects because there is not enough feature fusion across scales.

The contributions of this paper are as follows:An improved Triplet attention mechanism module is employed in the residual block of the backbone network to determine the importance of spatial and channel dimensions. This module improves the detection accuracy of the neural network, without a significant increase in computational effort.We improved the fusion mode of feature pyramids in the neck network by replacing PANet with BiFPN based on YOLOv4 to obtain richer localization and feature information.Compared to the original YOLOv4 model, the mean average precision (mAP) of this paper was improved by 8% for traffic sign detection on the TT100K-COCO dataset when the same input was used.

The rest of this paper is organized as follows. Section 2 introduces related works about object detection methods, attention mechanisms, and small object detection. Section 3 describes the proposed method in detail. The experimental results are presented in Section 4. Finally, Section 5 offers the conclusion.

## 2. Related Work

### 2.1. Object Detection Methods

Object detection, as one of the essential sub-tasks of computer vision, requires the correct identification of predefined classes in an image and the precise localization of their coordinates. Object detection is divided into one-stage [8,9,10,11,12] and two-stage [13,14,15,16] detection methods. The difference between the two is the presence or absence of the interest extraction step. The former can obtain faster inference, and the latter’s inference is based on the region of interest to get higher accuracy. The most typical one-stage detection methods are anchor-based YOLOv3 [9] and SSD [12], which can be predicted directly by the feature map. The principle is to predefine multiple anchor boxes for each cell of the feature map to detect the object, and then calculate the Intersection over Union (IOU) between anchor boxes and ground-truth boxes to delete the invalid and low-scoring anchor boxes. RCNN [13] uses a candidate region selection algorithm to obtain the input image containing the object, which is then used as the input to the convolutional network. However, feature region extraction can also be performed on the feature map. FasterRCNN [15] improves the inference speed of the network by completing the extraction of interest regions during the training process according to the Region Proposal Network (RPN) network.

Anchor-free one-stage networks [17,18,19] have been developed in recent years. Tian et al. [17] analyzed the disadvantages of predefined anchor boxes: (1) different hyperparameters need to be trained to adjust the deflation ratio when processing different datasets, and (2) there is a greater computational effort, which increases the burden on the hardware. To solve the above problems, the FCOS network was proposed, aiming to replace the anchor box with the center of the object and introduce centeredness to suppress the bounding box far from the center of the object to achieve a detection method that is not weaker than the anchor-based performance. The analysis of Zhang et al. [18] found that the differences between anchor-based and anchor-free performance stemmed from the selection of positive and negative samples, which were not significantly different under the same conditions, and proposed the Adaptive Training Sample Selection (ATSS) to classify positive and negative samples.

### 2.2. Attention Mechanism

The attention mechanism is often introduced as a module in backbone networks, such as [20,21,22]. SENet [20] is a channel attention mechanism that compensates for the shortcomings of convolutional networks focusing on local information by using the squeeze module to compress the feature map to obtain the global information, then using the excitation module to excite the channels containing important information, and finally applying the obtained excitation weights to the original feature map. CBAM [21] takes into account the spatial information while obtaining channel attention. First, it performs maximum global pooling and mean pooling on the input by channel to obtain spatial attention weights and apply them to the input feature layer. Then, it performs global pooling and mean pooling on the input by spatial to obtain spatial attention weights. The Triplet attention mechanism [23] is divided into three branches. The first two branches allow the channel dimension to respectively interact with the width and height dimensions, and the third branch allows the width and height dimensions to interact.

### 2.3. Small Object Detection

The detection of traffic signs plays a crucial role in driverless technology, guiding cars to follow traffic guidelines and ensuring driving safety. However, traffic signs occupy only a tiny portion of the area relative to the large and complex background, which places higher demands on the recognition and detection capabilities of network models. Most of the initial work focused on multi-scale image pyramids, such as SNIP [24] and SNIPER [25]. SNIP is a scale normalization method, which aims to solve the problem of small objects that are difficult to identify in the training sample by setting a range in advance. When things satisfy the range condition in images with different scales of the image pyramid, they are put into training, and those objects that are too small scale are eliminated to improve the accuracy of the network model. Since SNIP trains on all pixel points of the whole image, it leads to slow speed and requires more computational resources. To address the above shortcomings, SNIPER crops the image according to the regions in which the objects are located, but it leads to insufficient learning of background information, which increases the error rate of prediction results. Based on this, the false detection rate can be reduced by cropping the negative sample region, which does not contain the positive sample region containing the objects.

Since these two methods are based on image pyramids, which consume a lot of computational resources, feature pyramid structure networks [26,27,28,29] subsequently emerged. These networks often act as neck components. They are combined with the backbone network for feature fusion at different scales to obtain better prediction results. FPN [26] incorporates the low semantic features of the high-resolution feature map, and also combines the high semantic features of the low-resolution feature map, combining the more accurate localization information of the former and the greater amount of feature information of the latter, which greatly improves the detection accuracy. PANet [27] is improved for FPN networks by designing a top-down module to obtain richer features. BiFPN [28], as an improvement of the PANet network, optimizes the route of feature fusion by eliminating some unnecessary connections. Meanwhile, it constructs cross-layer connections and repeats the module as the basic unit of feature fusion several times. TridentNet [29] uses three branches to assign different dilation to the convolutions that need to be replaced, obtaining different ranges of receptive fields. Similar to SNIP, it uses the Regions of Interest (ROI) to perform pretraining. Those that meet the scope of candidate anchor boxes are considered positive samples and put into training; negative samples are ignored without back-propagation of the gradient.

## 3. Method

The YOLOv4 network structure has obtained good results in the field of object detection, and our proposed network is further optimized based on YOLOv4, mainly by improving the backbone network and the neck network. The overall structure of the improved network is shown in Figure 1.

### 3.1. Backbone

CSPDarknet53 is chosen as the backbone network. The nature of convolution is to make the network pay more attention to the local information, but the global and spatial information are ignored. Therefore, we combined the Triplet attention mechanism in the backbone network to pay more attention to global information. The specific location is shown in Figure 2.

The Triplet attention mechanism is a spatial and channel attention mechanism which mainly consists of three branches and focuses more on cross-dimensional connections. Each branch input is subjected to a *Z-pool* operation, which stacks the results after maximum global pooling and average pooling, as shown in Figure 3.

In the first branch, the input dimension is first adjusted to advance the width dimension, and the feature layer after *Z-pool* is convolved using a 7×7 convolution kernel, and then the weights are obtained using the sigmoid activation function. Finally, the weights are applied to the original input feature layer so that the feature layer focuses more on the dependence between height and channel dimension. If given a tensor, $x\in R^{C\times H\times W}$, where *H* and *W* are the spatial dimensions and *C* is the channel dimension. After that, the tensor *x* dimensionally adjusted tensor is denoted as $x_{1}$, which has the shape (W×C×H). After the *Z-pool* operation on $x_{1}$, the shape is (2×C×H), and after performing the 7×7 convolution, the tensor of shape (1×C×H) is obtained denoted as $\hat{x_{1}}$. In the next step, batch normalization is performed, and then the sigmoid activation function is used to obtain weights, which are then applied to $x_{1}$. Finally, the dimension is recovered as (C×H×W).

The second branch is similar to the first branch. The difference is that the height dimension is advanced and the shape is (H×C×W) noted as $x_{2}$. Finally, the weights of the channel and width are obtained, and applied to $x_{2}$. The dimension is recovered to (C×H×W).

In the last branch, the dimension is not adjusted, and is kept as it is, i.e., $x\in R^{C\times H\times W}$. After the *Z-pool* operation, we use convolution, batch normalization, and finally obtain the weights using the sigmoid activation function to learn the attention weights of height and width for application on *x*. The three branches are merged after completing their respective tasks, as shown in Equation (Equation 1):(1)$$O=\omega (x_{1}\sigma \left(\omega_{1}\left(\hat{x_{1}}\right)\right)+x_{2}\sigma \left(\omega_{2}\left(\hat{x_{2}}\right)\right)+x\sigma \left(\omega_{3}\left(\hat{x}\right)\right))$$
where $x_{1}$, $x_{2}$, *x* represent the inputs of three different branches, respectively, and these tensors are represented as $\hat{x_{1}}$, $\hat{x_{2}}$, $\hat{x}$ by a 7×7 convolution after *Z-pooling*. Meanwhile, σ represents sigmoid activate function and ω represents a learnable weight to optimize the fusion of the three branches.

### 3.2. Neck

For the model neck, we use BiFPN to replace PANet. BiFPN, as an improvement of PANet, can fuse more features across scales, as shown in Figure 4. First, the first change is that the input of the feature-enhanced network is changed from three to five, which means that the information of five different scale feature layers can be acquired. To facilitate the fusion, the number of channels of all inputs is uniformly adjusted to 256 before entering the BiFPN. Second, the connection routes of PANet feature layers are optimized. The $p_{1}$ feature layer should have existed between $p_{1}^{in}$ and $p_{1}^{out}$. $p_{1}$ has only one input compared to $p_{2}$, $p_{3}$, and $p_{4}$ which have two inputs and contain less feature information, so this is omitted to save computational resources. The $p_{5}$ feature layer consists of two inputs, $p_{4}$ and $p_{5}^{in}$, but $p_{5}^{out}$ is also composed of the same input, resulting in functional duplication, so $p_{5}$ is also omitted. In addition, three cross-layer connections are also established with the purpose of preventing the deeper feature layers from being feature agnostic, so that the deeper features are not weaker than the shallow ones. Most of the previous neck networks used to fuse information of different dimensions treat the contribution of each feature layer as the same, but each feature layer does not fill the same role. To make each component occupy the appropriate weight, the weights are therefore optimized, so that critical information occupies a greater weight and feature layers with less contribution occupy a lower weight. This module applies weights to each input and makes feature fusion more efficient via weighting. The formula is as follows:(2)$$Y=\sum_{k}\frac{w_{k}}{\varepsilon +\sum_{i}w_{i}}\cdot X_{k}$$
where $X_{k}$ represents input feature k, and $w_{k}$ represents a learnable weight for k. $w_{i}$ denotes all the weights of input tensors and *Y* denotes the output after feature fusion. A small value of ε is introduced to maintain stability, generally set to 0.0001.

The idea of this scheme is derived from soft-max, where the probability between 0 and 1 is obtained by dividing the total weight by the weight of each feature layer to limit the range of the weight change, which makes the gradient change more stable during the training process. Still, the exponential operation increases the amount of operation and slows down the GPU operation. To solve this problem, the exponential operation is therefore abandoned, and the total weights are divided by the weights directly, which can still obtain the probabilities in the 0-1 interval and improve the computational efficiency. It is worth noting that the weights are not constantly greater than 0, so the weights here must be limited to positive numbers.

Taking $p_{4}$ and $p_{4}^{out}$ as examples, we can see that the $p_{4}$ feature layer has two inputs, $p_{4}^{in}$ and ${\left(p_{3}\right)}^{\prime }$, after $p_{3}$ downsampling, while $p_{4}^{out}$ consists of three inputs: ${\left(p_{5}\right)}^{\prime }$ upsampled from $p_{5}^{out}$, $p_{4}$, and $p_{4}^{in}$. The formula is as follows:(3)$$p_{4}=\psi \left(\frac{w_{p_{3}}^{\prime }{\left(p_{3}\right)}^{\prime }+w_{p_{4}^{in}}p_{4}^{in}}{\varepsilon +w_{p_{3}}^{\prime }+w_{p_{4}^{in}}}\right)$$
(4)$$p_{4}^{out}=\psi \left(\frac{w_{p_{4}}^{\prime }p_{4}+w_{p_{4}^{in}}^{\prime }p_{4}^{in}+w_{\left(p_{5}\right)}^{\prime }{\left(p_{5}\right)}^{\prime }}{\varepsilon +w_{p_{4}}^{\prime }+w_{p_{4}^{in}}^{\prime }+w_{\left(p_{5}\right)}^{\prime }}\right)$$
where ψ donates depthwise separable convolution operation. In the top-down path, $p_{4}$ is the intermediate feature at layer 4, while in the bottom-up path, $p_{4}^{out}$ is the output feature at layer 4. $w_{p_{3}}^{\prime }$, $w_{p_{4}^{in}}$, $w_{p_{4}^{in}}^{\prime }$, $w_{\left(p_{5}\right)}^{\prime }$ represent respective weights.

## 4. Experiment Results

### 4.1. Dataset and Experimental Details

The experiment is run on a Linux server with Ubuntu 18.04, Intel(R) Xeon(R) CPU E5-2678 v3 @ 2.50 GHz, Tesla V100 GPU with 32 GB memory, based on the Pytorch deep learning framework, Pytorch version 1.11.0, CUDA version 11.6.

TT100K [30] consists of 100,000 images with 2048×2048 resolution, covering most traffic signs, which can provide support for both classification and recognition tasks of traffic signs in real-world scenes. The TT100K dataset contains objects of different sizes, divided into three sizes: small, medium, and large. The resolution of small objects is less than 32×32, the resolution of medium objects is between 32×32 and 96×96, and the rest are large objects. The 45 classes and the traffic lights and stop signs in the COCO [31] dataset are used as our dataset, called TT100K-COCO. TT100K-COCO contains a total of 9865 images, of which 8878 are used for the training process. The training set includes 7990 images, the validation set contains 888 images, and 987 images are used as tests. Due to the random division of images, the number of instances of classes varies greatly, which tests the detection ability of the model even more. The composition of the data set was shown in Table 1 and Figure 5.

### 4.2. Analysis of Results

To demonstrate the performance of the convolutional neural network proposed in this paper, some evaluation metrics are used to compare the SOTA models. Precision, which is the percentage of the number of correctly classified samples in the total sample, is shown as (Equation 5). Recall evaluates the ability of the model to find all positive samples, shown as (Equation 6). Therefore, we use mAP, which can reflect both precision and recall performance comprehensively, as an evaluation metric, shown in Equation (Equation 7). mAP@0.5 is the mAP value when the Intersection over Union (IoU) threshold is 0.5.
(5)$$\;\mathrm{Precision}\;=\frac{\mathrm{TP}}{\mathrm{TP}+\mathrm{FP}}$$
(6)$$\;\mathrm{Recall}\;=\frac{\mathrm{TP}}{\mathrm{TP}+\mathrm{FN}}$$
where TP (True Positive) means that the model predicts positive classes as positive, FP (False Positive) means that the model predicts negative classes as positive, and FN (False Negative) denotes that the model predicts positive classes as negative.
(7)$$\mathrm{AP}=\int_{0}^{1}p\left(r\right)dr$$
(8)$$\mathrm{mAP}=\frac{1}{\;N}\sum_{i}^{N}{\mathrm{AP}}_{i}$$
where AP is average precision and *p* and *r* are the values on the precision–recall curve, respectively. mAP means average AP for all classes.

As shown in Table 2, the mean average precision value of the original YOLOv4 model (mAP@0.5) is 52.6% on the test set, and the mAP of our improved model reaches 60.6%, which is an 8% improvement and achieves good results. In order to more accurately evaluate the detection ability of the model for different sizes of objects, the same evaluation metrics of the coco benchmark are used to calculate the mAP values for different sizes. The improved model achieves 14.8%, 39.4%, and 44.1% in small, middle, and large sizes, respectively, which is 2.3% higher, 2.2% higher, and 3.1% lower than the original model. Based on these results, we can conclude that our model has better accuracy than the original model in realistic scene detection filled with small and middle-sized targets, although it degrades in large targets. Compared to YOLOv3, the improvement is 2.4%, 10.3%, and 15.2%, respectively. Our model is further enhanced in small object detection when the input is 512 × 512, with a 3.3% improvement compared to 416 × 416.

Table 3 demonstrates the effect of the BiFPN number on FLOPs (Floating Point Operations), Parameters, and mAP. As the number of BiFPN module increases, the mAP improves on the TT100K-COCO dataset, even though the number of parameters and FLOPs also increases. That is, more computing resources are exchanged for the improvement in accuracy. Due to this, we can choose the number of BiFPNs to meet our needs according to the actual hardware situation.

Figure 6 shows the Miss Rate (MR) of each model at 16 different traffic signs. MR is used to indicate the rate of missed detection in the test results, as shown in Equation (Equation 9):(9)$$\mathrm{MR}=\frac{\mathrm{FN}}{\mathrm{TP}+\mathrm{FN}}$$
According to Equation (Equation 9), MR is actually the proportion of samples that were not correctly detected to the total number of samples. From Figure 6, it can be found that in general, our methods have a low MR, although some categories are higher.

Figure 7 shows the comparison of traffic sign detection results in different scenes. Figure 7a shows the original images, Figure 7b–d show the result images of YOLOv4, our method (416×416), and our method (512×512) detection, respectively. The detection of the first image shows that our model improves the detection of “ip” object by 5% and 13%, respectively, compared to the original YOLOv4. The detection of the second image is similar to the first one, with an improvement in accuracy. Then, we find that sample three has a smaller, darker, and more blurred object than the first two clearly visible samples, which directly leads to the missed detection of YOLOv4. At the same time, our method with input image 512×512 can detect the objects that are difficult to identify better than the model with input 416×416. In sample four, all three models detected the larger sign at the front, but the “pn” sign at the back was detected only by our model. Therefore, the method proposed by this paper is more adaptable and has significant performance for small objects, and especially has significant performance for small objects.

## 5. Conclusions

In this paper, we proposed an improved version of the algorithm based on YOLOv4 that focuses more on small object detection to overcome the challenges faced by traffic sign detection tasks in realistic scenarios. The following two practical and feasible improvements are proposed: an improved Triplet attention mechanism module used to make the fusion of the three branches more reasonable, and the use of a more nodal and structurally complex BiFPN instead of PANet to enhance cross-scale feature fusion. While our proposed algorithm achieves good results in the detection of small objects, there are still areas for improvement. In the future, the feature fusion network can be further optimized and streamlined to reduce the number of parameters, or the attention mechanism module could be added to this network. In the future, the feature fusion network may be further optimized and streamlined to reduce the number of parameters, or the attention mechanism module may be considered for addition to this network to improve the performance.

## Figures and Tables

**Figure 1 sensors-22-06930-f001:**
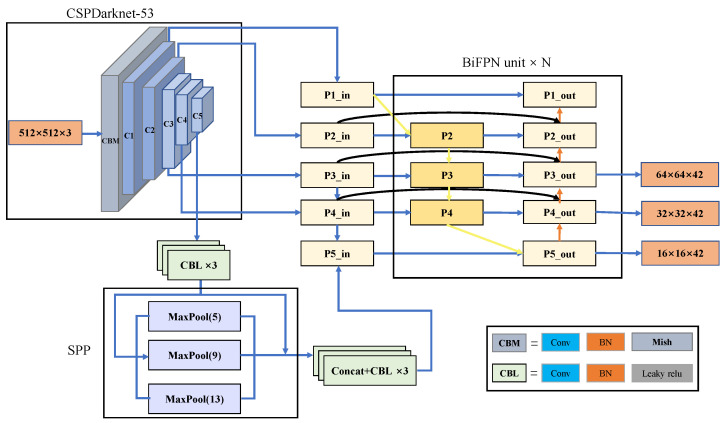
Architecture of improved YOLOv4.

**Figure 2 sensors-22-06930-f002:**
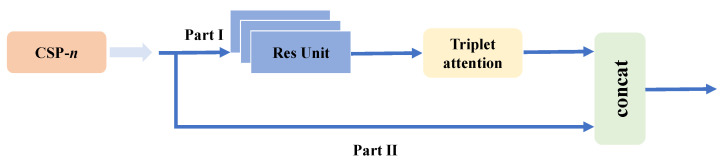
Triplet attention position.

**Figure 3 sensors-22-06930-f003:**
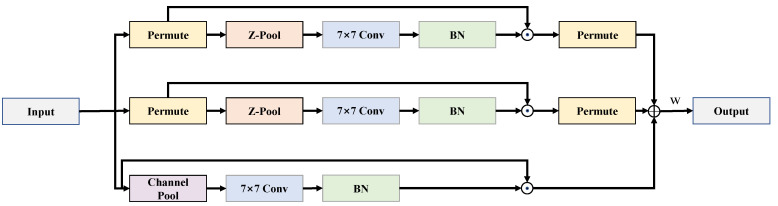
Architecture of improved triplet attention.

**Figure 4 sensors-22-06930-f004:**
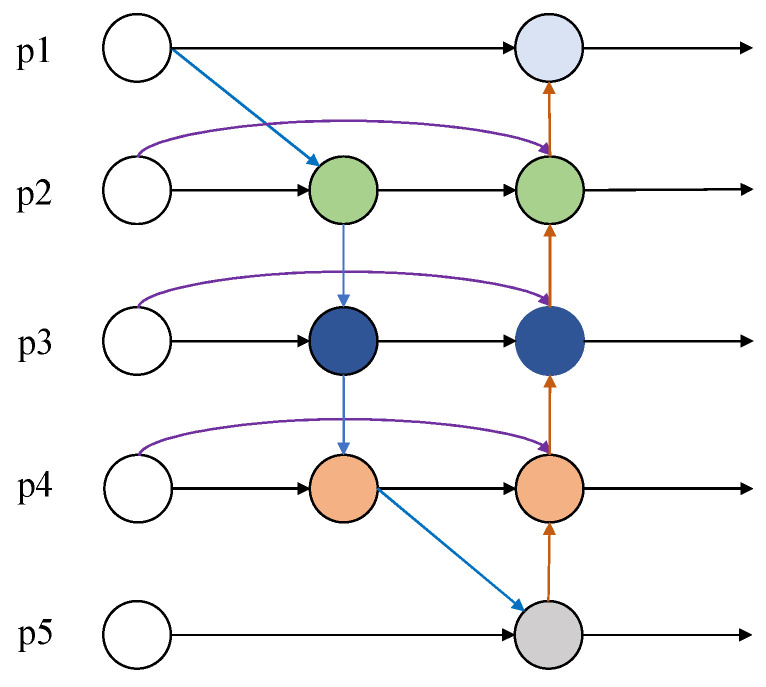
BiFPN Network.

**Figure 5 sensors-22-06930-f005:**
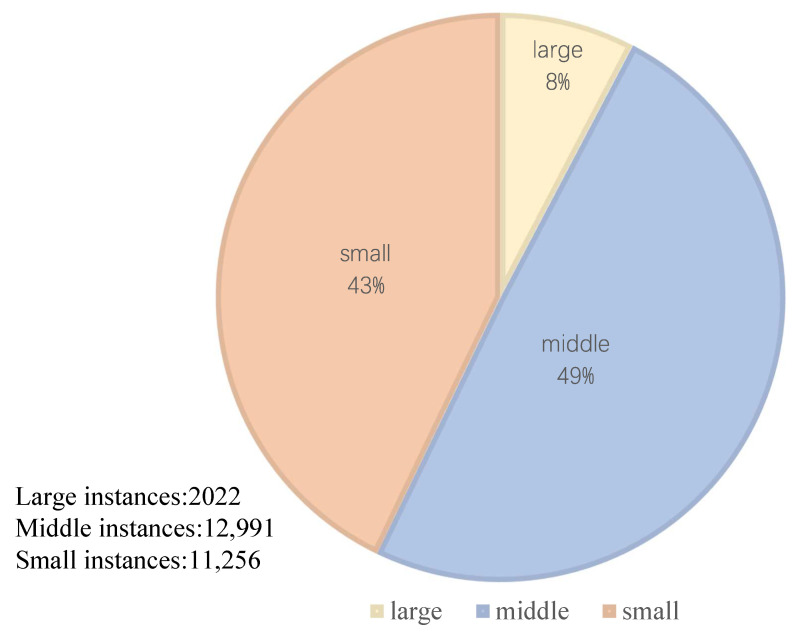
Instance size distribution.

**Figure 6 sensors-22-06930-f006:**
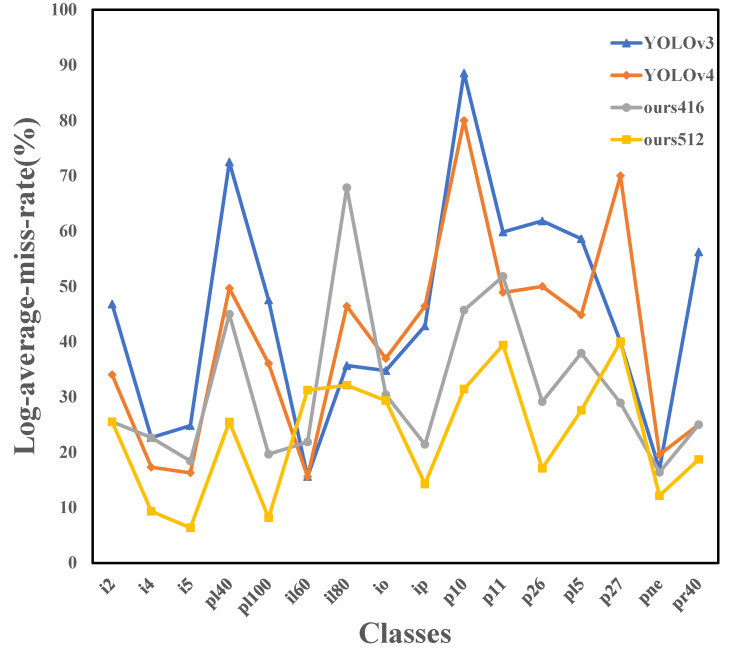
Comparison of miss rate.

**Figure 7 sensors-22-06930-f007:**
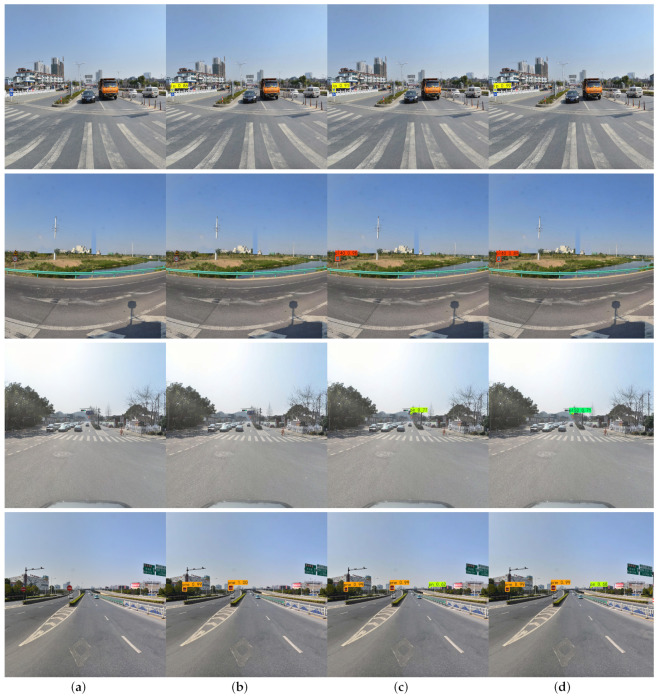
Comparison of traffic sign detection results.

**Table 1 sensors-22-06930-t001:** The composition of TT100K-COCO Dataset.

Classname	Instances	Classname	Instances	Classname	Instances	Classname	Instances
traffic light	1226	pl60	800	p11	1466	pm55	136
stop sign	338	w57	385	pl5	472	il80	293
pl40	1319	pl120	295	wo	111	w32	104
p26	756	pl100	665	io	846	il100	131
p27	131	il60	478	po	1124	p19	120
pne	2039	p10	331	i4	707	pr40	199
i5	1549	w55	169	pl70	147	ph4	120
p5	376	ph4.5	182	pl80	852	p23	266
ip	318	w13	121	pl50	1000	w59	180
pl30	578	pl20	154	i2	439	pm20	156
pn	2851	p12	172	pg	147	ph5	113
p3	139	p6	108	pm30	107	-	-

**Table 2 sensors-22-06930-t002:** mAP (%) obtained by several state-of-the-art methods.

Methods	Size	mAP@0.5	AP${}_{S}$	AP${}_{M}$	AP${}_{L}$
SSD [12]	300 × 300	34.2	2.9	19.1	58.1
YOLOv3 [9]	416 × 416	50.3	12.4	29.1	28.9
YOLOv4 [10]	416 × 416	52.6	12.5	37.2	48.3
Ours	416 × 416	60.6	14.8	39.4	44.1
Ours	512 × 512	66.4	18.1	42.6	45.2

**Table 3 sensors-22-06930-t003:** Results on different numbers of BiFPN modules.

Neck	FLOPs(G)	Parameters(M)	mAP@0.5
BiFPN × 1	58.56	56.676	55.88
BiFPN × 2	77.89	58.827	57.42
BiFPN × 3	97.23	60.977	60.6

## Data Availability

The TT100K-COCO dataset and codes are available on Github at https://github.com/waterdou/improved-yolov4.git, accessed on 8 August 2022.

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
