# Peer review of "Real-Time and Efficient Multi-Scale Traffic Sign Detection Method for Driverless Cars"

_sensors, 2022, doi:10.3390/s22186930_

Round 1

Reviewer 1 Report

Optional: The word 'tricks' in the abstract may be replaced by a more appropriate term. Suggestions: manipulations or further processing.

Author Response

Reviewer #1:

  1. The word 'tricks' in the abstract may be replaced by a more appropriate term.

Answer: We have modified a specific sentence.

Reviewer 2 Report

This paper optimizes the YOLOv4 network to improve the performance on detect small size objects. The triplet attention mechanism is applied in the residual block of the backbone network to capture cross-dimension features. Same as the replacement of PANet with BiFPN where the information feature can flow in both the upside and downside direction. The presentation of this paper is fluent, and the experiment analysis is sufficient. 

I have addressed several concerns as follows:

1. Is triplet attention applied to every residual block of the backbone network or all residual blocks using the same triplet attention module? 

2.  In lines 159, 165, 180, the convolution kernel is 7x7, and figure 3 shows as 3x3 conv.

3. In figure 6, the plot can have better visualization by using different marks for each benchmark.

Author Response

Reviewer :

  1. Is triplet attention applied to every residual block of the backbone network or all residual blocks using the same triplet attention module?

Answer: In fact, each residual block of the backbone network has its own triplet attention mechanism module, and the weights are not shared among them.

  1. In lines 159, 165, 180, the convolution kernel is 7x7, and figure 3 shows as 3x3 conv.

       Answer: We have modified the 3x3 convolution kernel in Figure 3 to 7x7.

  1. In figure 6, the plot can have better visualization by using different marks for each benchmark.

       Answer: To have a better visual effect, we assigned different markers to different networks.

Reviewer 3 Report

The paper reports on work that can improve signs detection for driverless car. A few comments for further improvement as below:-
1. Abstract - many acronyms / abbreviations which are undefined before use. Please define them properly. There is improper use of phrase (such as With some tricks), please be precise in stating your point. 

2. Table 2 - result analysis - please state the references in the table for the other 3 works, compared to your results.

3. Please justify on why mAP is a suitable figure of merit and mAP @ 0.5 is relevant in this context.

Author Response

Reviewer #2:

  1. Abstract -many acronyms / abbreviations which are undefined before use. Please define them properly. There is improper use of phrase (such as With some tricks), please be precise in stating your point.

Answer: We have defined several acronyms that were not previously defined and made “tricks” more specific.

  1. Table 2 - result analysis - please state the references in the table for the other 3 works, compared to your results.

Answer: We have added references in Table 2.

  1. Please justify on why mAP is a suitable figure of merit and mAP @ 0.5 is relevant in this context.

Answer: For mAP and mAP @ 0.5, we have briefly introduced what it is and what it does on page 8.